# Prognostic factors of first intimate partner violence among ever-married women in Sub-Saharan Africa: Gompertz gamma shared frailty modeling

**Beminate Lemma Seifu**[1]*, **Hiowt Altaye Asebe**[1], **Bruck Tesfaye Legesse**[2], **Getahun Fentaw Mulaw**[3,4], **Tsion Mulat Tebeje**[5], **Kusse Urmale Mare**[6]

1 Department of Public Health, College of Medicine and Health Sciences, Samara University, Samara, Ethiopia, 2 Department of Pediatrics and Neonatal Nursing, Institute of Health Sciences, School of Nursing and Midwifery, Wollega University, Nekemte, Ethiopia, 3 School of Pharmacy and Medical Sciences, Griffith University, Gold Coast, QLD, Australia, 4 Department of Public Health, College of Health and Medical Sciences, Woldia University, Woldia, Ethiopia, 5 School of Public Health, College of Health Sciences and Medicine, Dilla University, Dilla, Ethiopia, 6 Department of Nursing, College of Medicine and Health Sciences, Samara University, Samara, Ethiopia

* beminetlemma1915@gmail.com

**Data Availability Statement:** All DHS data files are publicly available from the http://www.dhsprogram.com database.

## Abstract

### Background

Violence against women, particularly intimate partner violence, is a significant Concern for public health as well as a violation of the human rights of women especially in low and middle-income countries. However, there was limited evidence how soon an ever-married women experience intimate partner violence in Africa. Therefore, this study aimed to investigate the timing of first intimate partner violence (FIPV) among ever-married women in 30 SSA countries and to identify the risk factors of the timing.

### Methods

The present study has utilized 125,731 weighted samples, who participated in the domestic violence module of the survey from Demographic and Health Surveys of 30 SSA countries. The Gompertz gamma shared frailty model was fitted to determine the predictors. For model evaluation, the theta value, Akaike Information Criteria (AIC), Bayesian Information Criteria (BIC), and deviance were used. The Adjusted Hazard Ratio (AHR) with a 95% Confidence Interval (CI) was reported in the multivariable Gompertz gamma shared frailty model to highlight the strength and statistical significance of the associations.

### Result

One-third (31.02%) of ever-married women had reported experiencing IPV. The overall incidence rate of FIPV was 57.68 persons per 1000 person-years (95% CI = 50.61–65.76). Age at marriage, age difference, educational status, employment, residence, women's decision-

**Funding:** The authors received no specific funding for this work.

**Competing interests:** The authors have declared that no competing interests exist.

**Abbreviations:** AHR, Adjusted Hazard Ratio; AIC, Akaike Information Criteria; CI, Confidence Interval; DHS, Demographic Health Survey; FIPV, First Intimate Partner Violence; IPV, Intimate Partner Violence; LLR, Log-likelihood Ratio; PH, Proportional Hazard; SSA, Sub-Saharan Africa; WHO, World Health Organization.

making autonomy, husband who drink alcohol and wealth status were significantly associated with the timing of FIPV.

## Conclusion

The findings show that ever-married women are at high and increasing risk of violence. Thus, we recommend establishing effective health and legal response services for IPV, strengthening laws governing the sale and purchase of alcohol, empowering women, raising the educational attainment of women, and putting policies in place to combat the culture of societal tolerance for IPV all contribute to the empowerment of women.

## Background

Intimate partner violence (IPV), as defined by the World Health Organization (WHO), is "behavior by an intimate partner or ex-partner that causes physical, sexual or psychological harm, including physical aggression, sexual coercion, psychological abuse and controlling behaviors" [1]. Women's violence, particularly IPV, is a significant public health concern as well as a violation of women's human rights especially in low and middle income countries (LMICs) [1].

Regardless of policies and interventions [2], women of all ages in LMICs continue to face several forms of violence, the most common of which is violence committed by an intimate partner [3,4]. A study conducted by WHO across 161 countries, found that worldwide, nearly 1 in 3, or 30%, of women have been subjected to physical and/or sexual violence by an intimate partner [5]. In the Sub- Saharan African (SSA) region, lifetime intimate partner violence prevalence estimates (since age 15) were 33%, ranging from 16% in Comoros to 47% in the Democratic Republic of Congo [6].

Violence against women is a global public health crisis that has serious social and economic consequences for countries as well as societies [6]. For women, physical, sexual, and psychological intimate partner violence can have substantial short- and long-term effects on their physical, mental, sexual, and reproductive health. It also has an impact on the health and well-being of their children [1]. These consequences can emerge as poor health and a low quality of life [7]. Women who were IPV survivors are more likely to have been injured in the head, face, neck, thorax, breasts, and abdomen than women who were injured in other ways [8]. Differential symptoms and conditions include sexually-transmitted diseases, vaginal bleeding or infection, fibroids, decreased sexual desire, genital irritation, pain on intercourse, chronic pelvic pain, and urinary-tract infections are the most consistent and long lasting gynecological problems among women who have history of IPV [9–11]. Besides the gynecological problems, researchers have found a significant relationship between mother's IPV experience and miscarriage [12–15]. Furthermore, Depression and post-traumatic stress disorder are the most common mental-health consequences of IPV, with significant comorbidity [16,17].

Studies on IPV showed that age of the woman, age at first marriage [18,19], educational status of the woman [19], woman's employment [18], residence [18,20], husband's educational status [21,22], household wealth status [22], women's household decision making autonomy [19,22], husband's age, husband's alcohol consumption [18,19,23], and parental history of spousal violence [19,22–24] were significant predictors of IPV.

Despite the fact that, the timing of the first intimate partner violence (FIPV) after a marriage has been investigated in SSA [18,24], Yet, those studies contain methodological flaws. Although the data from the demographic and health survey (DHS) are hierarchical, prior

research did not take into account the clustering effect or use the correct method of analysis, the frailty model (random effect survival model), to determine whether there was unobserved heterogeneity or shared frailty. Consequently, to account for the heterogeneity and to examine random variables that varies over the population, frailty model need to be carried out. Additionally, only 4 and 14 SSA nations were considered in these studies [18,24].

Therefore, this study aimed to investigate the timing of first intimate partner violence (FIPV) among ever-married women in 30 SSA countries and to identify the risk factors of the timing by using the appropriate statistical analysis method.

## Methods

### Data source and sampling procedure

The Demographic and Health Surveys' domestic violence module, which were carried out in 30 SSA countries between 2012 and 2021, provided the information used in this study. Over the past three decades, the DHS, a representative cross-sectional study carried out every five years, has served as a trustworthy source of data on a number of population health issues in developing countries. Using a two-stage stratified cluster sampling technique, households were chosen, and data were collected from women between the ages of 15 and 49 [25]. Among all eligible women in the household chosen for the individual questionnaire for this module, one woman was selected at random for the domestic violence module's individual questionnaire [25]. A total weighted sample of 125,731 ever married or cohabited women was included in the study (Table 1).

### Variables definition and measurement

To ascertain whether an ever-married woman experienced intimate partner violence prior to the survey dates, selected women respondents were asked the following questions

i. *Ever been slapped by husband/partner*

ii. *Ever been punched with fist or hit by something harmful by husband/partner*

iii. *Ever been kicked or dragged by husband/partner*

iv. *Ever been strangled or burnt by husband/partner*

v. *Ever been threatened with knife/gun or other weapon by husband/partner*

vi. *Ever CS physical violence by husband/partner*

vii. *Ever had arm twisted or hair pulled by husband/partner*

viii. *Ever been physically forced into unwanted sex by husband/partner*

ix. *Ever been forced into other unwanted sexual acts by husband/partner*

x. *Ever been physically forced to perform sexual acts respondent didn't want to*

Any woman who had ever been married and answered yes to at least one of the questions was considered to have experienced domestic violence. To determine the timing of first act of intimate partner violence women was then asked 'How long after you first (got married/started living together) with your (last) (husband/partner) did (this/any of these things) first happen?' [25].

The outcome variable was the time interval (survival time) between marriage date and the date of FIPV for those with reported violence. Women who had FIPV up until the survey date

**Table 1. Sample size of Demographic and health surveys and prevalence of intimate-partner violence across 30 Sub-Saharan African countries.**

| Country | Weighted frequency | Percentage | Prevalence of IPV (%) |
|---|---|---|---|
| Angola | 7,041 | 5.60 | 32.81 |
| Burkina Faso | 9,214 | 7.33 | 11.11 |
| Benin | 3,852 | 3.06 | 22.02 |
| Burundi | 6,441 | 5.12 | 46.82 |
| Democratic republic Congo | 4,692 | 3.73 | 47.76 |
| Cote di viorie | 4,281 | 3.41 | 24.56 |
| Cameroon | 4,092 | 3.25 | 34.71 |
| Ethiopia | 4,241 | 3.37 | 26.41 |
| Gabon | 2,531 | 2.01 | 43.05 |
| Gambia | 1,507 | 1.20 | 31.05 |
| Kenya | 3,966 | 3.15 | 39.12 |
| Comoros | 1,828 | 1.45 | 4.22 |
| Liberia | 1,716 | 1.37 | 42.20 |
| Madagascar | 5,222 | 4.15 | 26.83 |
| Mali | 2,928 | 2.33 | 37.12 |
| Mauritania | 2,366 | 1.88 | 8.53 |
| Malawi | 4,865 | 3.87 | 33.42 |
| Mozambique | 4,992 | 3.97 | 30.85 |
| Nigeria | 8,094 | 6.44 | 20.49 |
| Namibia | 882 | 0.70 | 21.17 |
| Rwanda | 1,680 | 1.34 | 39.87 |
| Serra Leone | 3,202 | 2.55 | 47.17 |
| Senegal | 1,088 | 0.87 | 13.97 |
| Chad | 3,214 | 2.56 | 26.06 |
| Togo | 4,729 | 3.76 | 21.04 |
| Tanzania | 7,008 | 5.57 | 41.11 |
| Uganda | 6,669 | 5.30 | 46.40 |
| South Africa | 1,902 | 1.51 | 4.39 |
| Zambia | 6,323 | 5.03 | 39.77 |
| Zimbabwe | 5,154 | 4.10 | 34.16 |

were regarded to have the event and coded as 1, while those who did not had the event were considered censored and coded as 0. Time at FIPV was recorded in years.

The independent variables considered for this study were categorized as socio-demographic, economic and behavioral variables. Such as women's age, age at first marriage, respondent's and husbands educational attainment, husband's age, respondent's and husband's employment, household wealth status, woman's household decision making autonomy, husband takes alcohol, age difference between respondent and husband, residence, and SSA region.

## Data management and analysis

Because of the non-proportional allocation to the sampling strata and the fixed sample size per cluster, all data were weighted to achieve acceptable representativeness at the national, regional, and county levels. For the statistical analysis, Stata 17 was employed.

The global Schoenfeld residuals test were used to check the proportional hazard (PH) assumption. We used a log-rank test with the null hypothesis of no difference between two or

more survival distributions at any point in time to examine the equality of the survival curve across population groupings.

Because the DHS data structure was hierarchical, we have checked whether there is clustering or not by running the frailty model (random effect survival model). The theta was significantly different from zero at the null model ($\theta = 0.12$, 95% CI: 0.09, 0.14) (LR test of theta = 0: $X^2 = 637.19$, $p < 0.001$). This shows that there is unobserved heterogeneity or a shared frailty, which means Women in a cluster had a higher likelihood of being correlated with other women in that cluster. Frailty models are used to account for unobserved heterogeneity or unmeasured factors that affect the hazard rate in survival analysis. These models introduce a random effect (frailty) that captures individual-specific variations not explained by observed covariates. Shared frailty models assume that the frailties are correlated across individuals within a cluster.

EAs/clusters were used as a random effect in a shared frailty model with baseline distributions (Weibull, Gompertz, and Exponential, log-logistic, and lognormal) and frailty distributions (gamma and inverse Gaussian). The Gompertz gamma shared frailty model got the highest log-likelihood and AIC values, making it the best-fitted model. In the multivariable analysis, variables with p-values less than 0.20 in the bivariable Gompertz gamma shared frailty analysis were included. The Adjusted Hazard Ratio (AHR) and 95% Confidence Interval (CI) were reported to declare the strength and significance of the associations between FIPV and independent variables in the multivariable Gompertz gamma shared frailty model.

## Ethical consideration

Data access permission was received from the measure DHS program via an online request at http://www.dhsprogram.com. The data used in this research were publicly available and did not contain any personal identifiers.

## Results

### Socio-demographic characteristics of ever-married women

Among the participants 87.01% of them were currently married, 81,659 (64.95%) of them were from rural residence, 39.79% were from poor households, whereas only 4.11% of them achieved higher educational level. Approximately two-third (69.56%) of the women has a decision-making autonomy and 68.50% of the women had a formal employment (Table 2).

### Prevalence and incidence rate of first intimate partner violence

In the 30 SSA countries, 125,731 ever-married women were sampled for domestic violence interviews, and 39,007 (31.02%; 95% CI = 30.8%-31.3%) of those women reported experiencing domestic violence after marriage. On average, women experienced the first intimate partner violence 3.39 years after union (95%CI: 3.35, 3.43).

The overall incidence rate of first intimate partner violence was 57.68 persons per 1000 person-years (95% CI = 50.61–65.76). The incidence of FIPV at the end of the first 5 years, 10 years, and 15 years, 20 years and above 20 years was 159.44, 30.30, 7.05, 6.85 and 1.19 per 1000 person-years, respectively.

### Comparison of failure functions

Visual comparison of the probability of IPV across categorical explanatory variables was done using the Kaplan-Meier (KM) failure curve. The overall KM failure curve showed that the likelihood of experiencing IPV increased with time (Fig 1). From the log-rank test, a statistically

**Table 2. Distribution of ever-married women by sociodemographic and economic characteristics.**

| Variable | Weighted frequency (n = 125,731) | Percentage (%) |
|---|---|---|
| **Age (years)** | | |
| 15–19 | 7,776 | 6.18 |
| 20–29 | 46,632 | 37.09 |
| 30–39 | 43,646 | 34.71 |
| 40–49 | 27,676 | 22.01 |
| **Currently/ formerly in union** | | |
| Currently in union/living with a man | 109,392 | 87.01 |
| Formerly in union/living with a man* | 16,338 | 12.99 |
| **Age at marriage** | | |
| <15 | 28,489 | 22.66 |
| 15–19 | 54,066 | 43.00 |
| 20–24 | 30,355 | 24.14 |
| >25 | 12,820 | 10.20 |
| **Woman's education** | | |
| No education | 45,318 | 36.04 |
| primary | 44,692 | 35.55 |
| Secondary | 30,549 | 24.30 |
| Higher | 5,171 | 4.11 |
| **Husband/partner's education** | | |
| No education | 34,907 | 30.81 |
| Primary | 35,598 | 31.42 |
| Secondary | 33,813 | 29.85 |
| Higher | 8,968 | 7.92 |
| **Residence** | | |
| Urban | 44,072 | 35.05 |
| Rural | 81,659 | 64.95 |
| **Employment** | | |
| Not employed | 39,584 | 31.50 |
| Employed | 86,079 | 68.50 |
| **Husband/partner's employment** | | |
| Not employed | 5,575 | 4.43 |
| Employed | 120,156 | 95.57 |
| **Household wealth index** | | |
| Poor | 50,029 | 39.79 |
| Middle | 25,199 | 20.04 |
| Rich | 50,502 | 40.17 |
| **Age difference (partner)** | | |
| Younger | 61 | 0.05 |
| Same age | 446 | 0.35 |
| Older | 125,224 | 99.60 |
| **Husband drinks alcohol** | | |
| No | 77,784 | 63.08 |
| Yes | 45,535 | 36.92 |
| **Women decision making autonomy** | | |
| No autonomy | 38,268 | 30.44 |
| Has autonomy | 87,462 | 69.56 |
| **Sub-Saharan Africa region** | | |

(*Continued*)

**Table 2.** (Continued)

| Variable | Weighted frequency (n = 125,731) | Percentage (%) |
|---|---|---|
| Central Africa | 23,069 | 18.35 |
| Eastern Africa | 28,937 | 23.01 |
| Southern Africa | 24,121 | 19.18 |
| Western Africa | 49,604 | 39.45 |

*divorced, widowed, separated.

significant difference existed in IPV across age, woman's education, husband education, woman's occupation, husband occupation, age difference, husband drinks alcohol, woman's decision making autonomy, wealth status, and SSA region (log-rank, p<005) (Table 3).

## Assessment of the proportional hazard assumption

The global Schoenfeld residuals test was used to examine the proportional hazard assumption for all potential determinants of intimate partner violence. When the global Schoenfeld residuals test yields a p-value less than 0.05, it indicates that the proportional hazards (PH) assumption has been violated. Consequently, relying solely on a proportional hazard model (PHM)

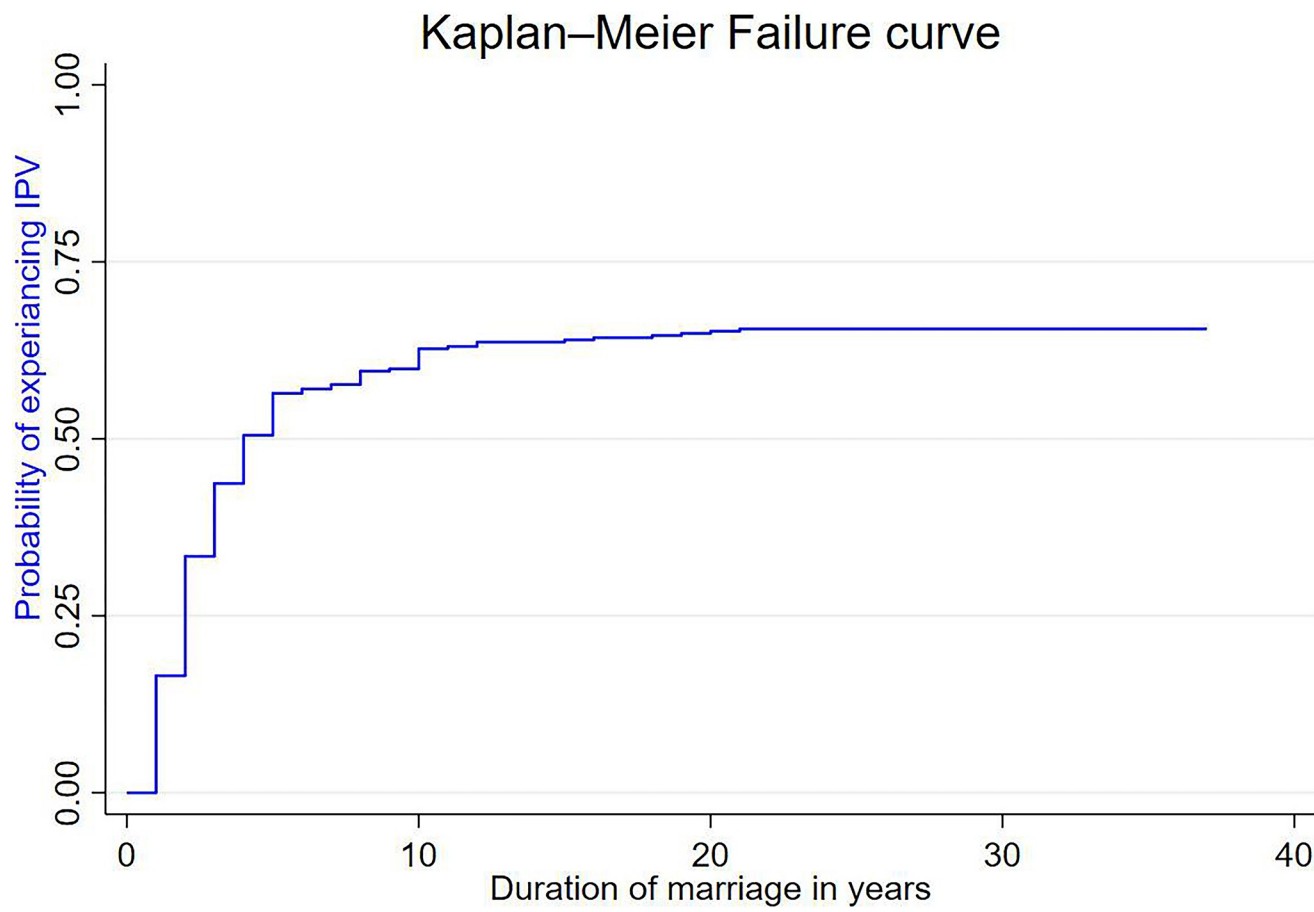

**Fig 1. The Kaplan-Meier failure curve of intimate partner violence in Sub-Saharan Africa.**

**Table 3. Log-rank test for the predicators of first intimate partner violence.**

| Variables | p-value | Variables | p-value |
|---|---|---|---|
| Age | <0.001 | Partner's employment status | 0.012 |
| Age at first marriage | <0.001 | Age difference | <0.001 |
| Women's educational status | <0.001 | Wealth status | 0.017 |
| Partner's educational status | <0.001 | Husband drinks alcohol | <0.001 |
| Woman's decision making autonomy | <0.001 | SSA region | 0.008 |

would not be appropriate in this scenario. The violation of the PH assumption implies that the hazard ratios for the covariates are not constant over time. In such cases, alternative survival models that account for time-varying effects or unobserved heterogeneity become necessary. Therefore, parametric survival models were fitted to get a reliable estimate (Table 4).

### Predictors of time to first intimate partner violence

Based on deviance, AIC, BIC, and theta value, the shared frailty with Gompertz distribution and gamma frailty was the best-fitted model for the data (Table 4). In the final model woman's age, women educational and employment status, residence, woman's decision-making autonomy, wealth status and husband drinks alcohol were significant predictors of intimate partner violence. Compared to women aged 15–19 years, women aged 20–29 and 45–49 had about 35% (AHR = 0.65, 95% CI: 0.61, 0.69) and 66% (AHR = 0.34, 95% CI: 0.32, 0.37) decreased hazard of experiencing IPV. Rural resident women were 6% less hazard (AHR = 0.94, 95% CI: 0.91, 0.97) to experience IPV. Women who had a formal employment had 1.14 times (AHR = 1.14, 95% CI: 1.11, 1.17) more hazard of experiencing IPV. Similarly women who had a primary and secondary educational achievement had 1.25 times (AHR = 1.25, 95% CI: 1.21, 1.29) and 1.22 (AHR = 1.22, 95% CI: 1.18, 1.27) increased hazard to be violated by their intimate partner compared to women who do not have a formal education, respectively (Table 5).

### Discussion

According to the current study's finding, in Africa, one out of every three women who have ever married has had experienced IPV in their lifetime. In the Gompertz gamma shared frailty

**Table 4. Schoenfeld residual test for checking the proportional hazard assumption for the intimate partner violence and its predicators among reproductive ever-married women in SSA.**

| Variables | Rho | Chi$^2$ | Df | Prob>chi$^2$ |
|---|---|---|---|---|
| Age | -0.03741 | 41.44 | 1 | 0.0001 |
| Age at first marriage | 0.04579 | 60.32 | 1 | 0.0001 |
| Women's educational status | 0.01192 | 3.93 | 1 | 0.0475 |
| Partner's educational status | 0.00079 | 0.02 | 1 | 0.8954 |
| Residence | 0.01231 | 4.33 | 1 | 0.0376 |
| Woman's employment status | 0.01614 | 7.38 | 1 | 0.0066 |
| Partner's employment status | -0.00066 | 0.01 | 1 | 0.9122 |
| Age difference | -0.01619 | 6.78 | 1 | 0.0092 |
| Wealth status | 0.02402 | 16.74 | 1 | 0.0001 |
| Husband drinks alcohol | 0.03885 | 43.05 | 1 | 0.0001 |
| Woman's decision making autonomy | 0.03793 | 41.72 | 1 | 0.0001 |
| SSA region | -0.04615 | 60.02 | 1 | 0.0001 |
| Global test | | 387.91 | 12 | 0.0001 |

**Table 5. The bi-variable and multivariable Gompertz gamma shared frailty model for predictors of the first Intimate partner violence in SSA.**

| Variable | Intimate partner violence | | Hazard Ratio (HR) with 95% CI | |
|---|---|---|---|---|
| | Censored | Event | CHR 95%% CI | AHR 95 CI% |
| **Age** | | | | |
| 15–19 | 5,905 | 1,871 | 1 | 1 |
| 20–29 | 31,723 | 14,909 | 0.73 (0.68, 0.77) | 0.65 (0.61, 0.69)* |
| 30–39 | 29,768 | 13,878 | 0.51 (0.48, 0.55) | 0.43 (0.40, 0.46)* |
| 40–49 | 19,327 | 8,349 | 0.41 (0.38, 0.44) | 0.34 (0.32, 0.37)* |
| **Age at first marriage** | | | | |
| <15 | 19,623 | 8,867 | 1 | 1 |
| 15–19 | 35,956 | 18,110 | 1.16 (1.13, 1.19) | 1.16 (1.12, 1.19)* |
| 20–24 | 21,282 | 9,073 | 1.06 (1.03, 1.09) | 1.16 (1.12, 1.20)* |
| >25 | 9,862 | 2,958 | 0.93 (0.89. 0.98) | 1.17 (1.11, 1.24)* |
| **Age difference with husband** | | | | |
| Younger | 41 | 20 | 1 | 1 |
| Same age | 264 | 182 | 1.91 (1.06, 3.44) | 1.78 (0.99, 3.21) |
| Older | 86,419 | 38,806 | 0.93 (0.52, 1.64) | 1.31 (0.75, 2.32) |
| **Education** | | | | |
| No education | 33,095 | 12,223 | 1 | 1 |
| Primary | 28,032 | 16,660 | 1.46 (1.42, 1.49) | 1.23 (1.19, 1.27)* |
| Secondary | 21,365 | 9,184 | 1.29 (1.25, 1.33) | 1.12 (1.08, 1.17)* |
| Higher | 4,231 | 941 | 0.71 (0.66, 0.77) | 0.79 (0.72, 0.88)* |
| **Husband education** | | | | |
| No education | 26,454 | 8,453 | 1 | 1 |
| Primary | 22,984 | 12,615 | 1.67 (1.62, 1.72) | 1.25 (1.21, 1.29)* |
| Secondary | 23,222 | 10,591 | 1.51 (1.47, 1.56) | 1.22 (1.18, 1.27)* |
| Higher | 7,037 | 1,931 | 1.01 (0.95, 1.06) | 1.03 (0.97, 1.11)* |
| **Employment** | | | | |
| Not employed | 29,057 | 10,527 | 1 | 1 |
| Employed | 57,616 | 28,463 | 1.26 (1.22, 1.29) | 1.14 (1.11, 1.17)* |
| **Husband employment** | | | | |
| Not employed | 3,937 | 1,638 | 1 | 1 |
| Employed | 82,787 | 37,369 | 1.07 (1.01, 1.13) | 0.97 (0.91, 1.03) |
| **Residence** | | | | |
| Urban | 31,058 | 13,014 | 1 | 1 |
| Rural | 55,665 | 25,994 | 1.08 (1.06, 1.11) | 0.94 (0.91, 0.97)* |
| **Household wealth status** | | | | |
| Poor | 33,618 | 16,411 | 1 | 1 |
| Middle | 17,180 | 8,020 | 0.95 (0.92, 0.98) | 0.92 (0.89, 0.94)* |
| Rich | 35,926 | 14,576 | 0.85 (0.83, 0.87) | 0.82 (0.79, 0.84)* |
| **Husband drinks alcohol** | | | | |
| No | 59,863 | 17,921 | 1 | 1 |
| Yes | 24,655 | 20,881 | 2.35 (2.29, 2.40) | 2.17 (2.11, 2.22)* |
| **Women decision making autonomy** | | | | |
| No autonomy | 25,590 | 12,678 | 1 | 1 |
| Has autonomy | 61,133 | 26,329 | 1.29 (1.26, 1.32) | 0.91 (0.88, 0.94)* |
| **Sub-Saharan African region** | | | | |
| Central Africa | 14,024 | 9,046 | 1 | 1 |
| Eastern Africa | 18,818 | 10,119 | 0.62 (0.60, 0.64) | 0.89 (0.87, 0.93)** |

*(Continued)*

**Table 5.** (Continued)

| Variable | Intimate partner violence | | Hazard Ratio (HR) with 95% CI | |
|---|---|---|---|---|
| | Censored | Event | CHR 95%% CI | AHR 95 CI% |
| Southern Africa | 16,499 | 7,622 | 0.70 (0.68, 0.73) | 0.74 (0.71, 0.77)* |
| Western Africa | 37,383 | 12,221 | 0.58 (0.57, 0.60) | 0.74 (0.71, 0.76)* |

AHR: Adjusted Hazard Ratio, CHR: Crude Hazard Ratio, CI: Confidence Interval.

*P-value ≤0.05.

model; Woman's age, age difference with husband/partner, woman's education, employment and wealth status, residence, decision making autonomy and husband drinks alcohol were the significant predictors of timing to the first intimate partner violence.

Among women who have been married, the average time until they experience abuse after marriage is approximately 3.4 years. Notably, 38.94% of these women encounter abuse within the first year, while 44.08% face it within the initial five years of their union. This is in line with the average of 3.5 years reported by a study conducted among 30 developing countries

According to our study women who had primary or secondary education and employed had a higher risk of IPV than women who do not have a formal education and employment, as ascertained in previous studies [18,26–28]. In patriarchal societies like SSA countries where male dominance over and violence against women are tolerated, neither women's education level, nor employment status are protective against IPV [29,30]. Due to the possibility that women's socioeconomic independence may be related to the shift in marital power dynamics being at odds with society norms and being regarded as undermining the position of the male [31,32]. As a result, violence is used to balance power within a relationship, and unequal access to resources relative to the other spouse, may prompt the use of violence to restore power superiority within a partnership [33,34].

In this study old women's age found to be a protective factor against IPV. This is in line with findings from previous studies [26,35,36]. The risks of IPV were higher among women whose husband/partner is of the same age or older than them. The fact that can be used to explain this finding is low relationship power often results from partner age disparity [37]. The degree to which one may act independently of a partner's control, influence a partner's activities, and dominate decision-making is referred to as relationship power; it covers the domains of relationship control and decision-making domination [38]. Low relational power has been connected to IPV [39,40].

The current study demonstrate that having a decision making autonomy is associated with reduced risk of having experienced intimate partner violence, this is consistent with studies done in Turkey [41], Bangladesh [42], and Peru [43]. This also supports the theory that women's decision-making autonomy in the household reduces the incidence of domestic violence. This could be because women who are empowered or have decision-making autonomy can fight for their rights and will not allow men entirely dictating to them, which could end in sexual, physical, or emotional violence [44].

In line with prior findings [23,45,46], When compared to their counterparts, the risk of IPV among women who had an alcoholic partner was higher. This could be related to the influence of alcohol on cognitive abilities, lowering self-control of individuals' lower inhibitions, and enhancing patriarchal ideas, thereby triggering authoritarian toxic masculinities which in turn encourages violent conduct [47]. Furthermore, alcohol usage has been linked to having multiple sexual partners [48], and excess alcohol spending can reduce family income and contribute to tension, which can lead to conflict and violence [49].

Previous research has found that a woman's wealth status may be a protective factor against IPV [50–53]. Similarly, in this study, women with middle and rich wealth statushad lower hazards of experiencing IPV than did women with poor wealth status. This could be due to wealth-driven empowerment, which eventually lessens their reliance on their partner, furthermore, poor women are heavily reliant on their relationships and may be unable to argue [51,52].

In this study, being rural resident was found to be a protective factor against IPV. This could be because women in rural areas are more inclined to accept IPV and do not report it, resulting in an underestimation of the incidence. According to studies, the entrenched and inflexible patriarchal culture in rural communities makes reporting incidents of violence practically impossible due to the perception that doing so would be disrespectful to the spouse and to family members and elders whose jobs involve mediating in such disputes [54,55].

## Study limitations and strengths

The study's findings must be considered in light of the following limitations. First, the study's cross-sectional design makes determining cause and effect relationships impossible. Second, all variables, including partner traits, were self-reported, raising the possibility of recall bias. Third, because these factors were not included in the dataset, some ecological model variables such as social support for victims, neighborhood environment, legislation, and national policies were not analyzed in this study. The data for this study came from a pooled nationally representative DHS survey of 30 Sub-Saharan African nations, which made the observed associations more robust and enhanced the generalizability. Furthermore, advanced statistical modeling that takes into account the nested nature of the DHS data was used to get credible standard error and estimate.

## Conclusion

Intimate partner violence is high among ever-married women in SSA. The timing of FIPV was associated to education, employment, whether the husband drinks alcohol, woman's decision-making autonomy and wealth quintile, as well as residence. The study's findings have a variety of ramifications for women's health promotion and protection. In the SSA context, intervention services need to target alcoholic husbands, empowering women, raising the educational attainment of women, and putting policies in place to combat the culture of societal tolerance for IPV all contribute to the empowerment of women. Furthermore, since the highest incidence of IPV among women is concerning, Policies as well as services focused on preventing and combating IPV should be expanded, and existing programs should be customized to women.

## Author Contributions

**Conceptualization:** Beminate Lemma Seifu, Hiowt Altaye Asebe, Getahun Fentaw Mulaw, Kusse Urmale Mare.

**Data curation:** Beminate Lemma Seifu, Bruck Tesfaye Legesse, Kusse Urmale Mare.

**Formal analysis:** Beminate Lemma Seifu, Bruck Tesfaye Legesse, Getahun Fentaw Mulaw, Kusse Urmale Mare.

**Methodology:** Beminate Lemma Seifu, Hiowt Altaye Asebe, Bruck Tesfaye Legesse, Getahun Fentaw Mulaw, Tsion Mulat Tebeje, Kusse Urmale Mare.

**Software:** Beminate Lemma Seifu, Hiowt Altaye Asebe, Bruck Tesfaye Legesse, Tsion Mulat Tebeje, Kusse Urmale Mare.

**Supervision:** Beminate Lemma Seifu, Kusse Urmale Mare.

**Validation:** Beminate Lemma Seifu, Bruck Tesfaye Legesse, Kusse Urmale Mare.

**Visualization:** Beminate Lemma Seifu, Hiowt Altaye Asebe, Bruck Tesfaye Legesse, Getahun Fentaw Mulaw, Tsion Mulat Tebeje, Kusse Urmale Mare.

**Writing – original draft:** Beminate Lemma Seifu, Hiowt Altaye Asebe, Bruck Tesfaye Legesse, Getahun Fentaw Mulaw, Tsion Mulat Tebeje, Kusse Urmale Mare.

**Writing – review & editing:** Beminate Lemma Seifu, Hiowt Altaye Asebe, Bruck Tesfaye Legesse, Getahun Fentaw Mulaw, Tsion Mulat Tebeje, Kusse Urmale Mare.

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
