## [Decision Letter · Decision Letter 0]

20 Nov 2023

PONE-D-23-11923Prognostic factors of first intimate partner violence among ever-married women in Sub-Saharan Africa: Gompertz gamma shared frailty modelingPLOS ONE

Dear Dr. Seifu,

Thank you for submitting your manuscript to PLOS ONE. After careful consideration, we feel that it has merit but does not fully meet PLOS ONE’s publication criteria as it currently stands. Therefore, we invite you to submit a revised version of the manuscript that addresses the points raised during the review process.

We look forward to receiving your revised manuscript.

Kind regards,

Obasanjo Afolabi Bolarinwa, Masters

Academic Editor

PLOS ONE

Journal Requirements:

Reviewers' comments:

Reviewer's Responses to Questions

**Comments to the Author**

1. Is the manuscript technically sound, and do the data support the conclusions?

Reviewer #1: Yes

Reviewer #2: Yes

2. Has the statistical analysis been performed appropriately and rigorously? 

Reviewer #1: Yes

Reviewer #2: Yes

3. Have the authors made all data underlying the findings in their manuscript fully available?

Reviewer #1: Yes

Reviewer #2: Yes

4. Is the manuscript presented in an intelligible fashion and written in standard English?

Reviewer #1: Yes

Reviewer #2: Yes

5. Review Comments to the Author

Reviewer #1: The manuscript is technically sound and uses fancy statistics but does not answer the research question, namely 'Therefore, this study aimed to investigate the timing of first intimate partner

violence (FIPV) among ever-married women in 30 SSA countries and to identify the

risk factors of the timing by using the appropriate statistical analysis method.'

Methods: The present study has utilized 125,731 weighted samples, BUT it does not say from which countries which is not helpful for targeted recommendations because countries differ.

Results. Nothing new. The problem has been rehashed many times. These statistics are not unknown. Perhaps if we understood better when in the marriage the violence was initiated and the circumstances, interventions could be better targeted. Thus the authors need to present 'the timing of first intimate partner

violence (FIPV) among ever-married women' and 'risk factors of the timing ' AND whether this is similar or how it differs in the 30 countries.

Discussion. THEN the discussion needs to address the research questions as above.

EXAMPLES OF ENGLISH CORRECTIONS REQUIRED.

Line 177. majority (64.95%) of them were from rural residence,

Line 178,, More than half (69.56%) of the women has a decision making autonomy

Reviewer #2: Arrange the "Key word" alphabetically (line 50)

The introduction was well written highlighting the issue and the gaps of previous studies that necessitate this study. The objective of the study is clear.

Method

The design of the study is clearly stated--DHS & cross-sectional. The outcome and independent variables was clearly spelt out and the methodology is well written

Discussion

was the statement in line 228-229 a result from this study? If yes, please recast to capture so.

What is the policy implication of this study?

6. PLOS authors have the option to publish the peer review history of their article (what does this mean?). If published, this will include your full peer review and any attached files.

Reviewer #1: No

Reviewer #2: **Yes: **Chukwudeh Okechukwu Stephen

---

## [Author Response · Author response to Decision Letter 0]

26 Nov 2023

Point-by-point response 

Point by point response for editors/reviewers comments 

PLOS ONE

Manuscript title: Prognostic factors of first intimate partner violence among ever-married women in Sub-Saharan Africa: Gompertz gamma shared frailty modeling 

Manuscript ID: PONE-D-23-11923

 Dear editor/reviewer. 

Dear all,

We would like to thank you for the constructive, building, and improvable comments on this manuscript that would improve the content of the manuscript. We considered each comment and clarification question of editors and reviewers on the manuscript thoroughly. Our point-by-point responses for each comment and question are described in detail on the following pages. Further, the details of changes were shown by track changes in the supplementary document attached

'Response to Reviewers

 Editor Comments:

Author’s response: Dear editor thank you for your comment. We have ensured that the manuscript is written according to PLOS ONE's style requirements, including those for file naming.

Author’s response: Dear editor thank you for your comment. Data access permission was received from the measure DHS program via an online request at http://www.dhsprogram.com . The data used in this research were publicly available and did not contain any personal identifiers. We have acknowledged this in the “ethics statement” of the method section of the manuscript. (please see the revised manuscript).

Author’s response: Thank you dear editor for your suggestion. The dataset used in this study is available online and can be accessed at www.measuredhs.com.

Author’s response: Thank you dear editor for your suggestion. We have verified that all the references cited in this study remain valid and have not been retracted. 

Response to reviewer 1 

Reviewer #1: The manuscript is technically sound and uses fancy statistics but does not answer the research question, namely 'Therefore, this study aimed to investigate the timing of first intimate partner violence (FIPV) among ever-married women in 30 SSA countries and to identify the risk factors of the timing by using the appropriate statistical analysis method.'

Author’s response: Dear reviewer thank you for your comment. We have answered our research question to ‘investigate the timing of first intimate partner violence (FIPV) among ever-married women in 30 SSA countries and to identify the risk factors of the timing by using the appropriate statistical analysis method.' Indeed, survival analysis is a powerful tool for addressing research questions that involve time-to-event outcomes. By considering the time component and estimating hazard ratios, we gain insights into the factors influencing the outcome, timing of the first intimate partner violence. Furthermore, we calculated the person-time-at-risk, which can provide us the information the rate of the FIPV or how quickly women are experiencing the FIPV following their marriage. 

Methods: The present study has utilized 125,731 weighted samples, BUT it does not say from which countries, which is not helpful for targeted recommendations because countries differ.

Author’s response: Dear reviewer than you for your comment. We have included a table representing the list of countries with their respective weighted frequency and proportion of intimate partner violence across the included countries. 

Results. Nothing new. The problem has been rehashed many times. These statistics are not unknown. Perhaps if we understood better when in the marriage the violence was initiated and the circumstances, interventions could be better targeted. Thus the authors need to present 'the timing of first intimate partner violence (FIPV) among ever-married women' and 'risk factors of the timing ' AND whether this is similar or how it differs in the 30 countries.

Author’s response: Dear reviewer thank you for your comments. Prior research has focused on intimate partner violence within individual SSA nations, examining significant predictors. In our current study, we aim to provide a broader SSA-level perspective by investigating the incidence and predictors of first intimate partner violence among all married women. Our use of a representative sample size ensures robust statistical power and relatively unbiased estimates. These findings hold valuable implications for policymaking and targeted interventions across SSA nations. When dealing with heterogeneity, we often use random effects models (frailty in our case). These models incorporate random effects (such as country-specific intercepts) to account for variability between different units (e.g., countries). 

The estimated median survival remains elusive, as more than half of the individuals have yet to encounter their first instance of intimate partner violence (FIPV). In this context, calculating a median survival is not feasible. However, we can get insights from the overall incident rate and the incidence rate of FIPV over 5-year intervals within marriages or unions. These rates provide valuable information about the occurrence and patterns of FIPV across different durations of relationships.

EXAMPLES OF ENGLISH CORRECTIONS REQUIRED.

Line 177. majority (64.95%) of them were from rural residence,

Line 178,, More than half (69.56%) of the women has a decision making autonomy

Author’s response: Thank you dear reviewer. We have corrected. 

Response to reviewer 2

Arrange the "Key word" alphabetically (line 50)

Author’s response: Dear reviewer thank you for your comment. We have arranged in ascending alphabetical order. 

The introduction was well written highlighting the issue and the gaps of previous studies that necessitate this study. The objective of the study is clear.

Author’s response: Thank you for your kind words! We appreciate your compliment.

Method

The design of the study is clearly stated--DHS & cross-sectional. The outcome and independent variables was clearly spelt out and the methodology is well written

Author’s response: Thank you for your kind words! We appreciate your compliment.

Discussion

was the statement in line 228-229 a result from this study? If yes, please recast to capture so.

Author’s response: Thank you dear reviewer for your concern. Yes, the statement was based on our finding and we have recast it.

---

## [Decision Letter · Decision Letter 1]

13 Mar 2024

PONE-D-23-11923R1Prognostic factors of first intimate partner violence among ever-married women in Sub-Saharan Africa: Gompertz gamma shared frailty modelingPLOS ONE

Dear Dr. Seifu,

Thank you for submitting your manuscript to PLOS ONE. After careful consideration, we feel that it has merit but does not fully meet PLOS ONE’s publication criteria as it currently stands. Therefore, we invite you to submit a revised version of the manuscript that addresses the points raised during the review process.

Please find editor's concerns and address them.

We look forward to receiving your revised manuscript.

Kind regards,

Enamul Kabir

Academic Editor

PLOS ONE

Additional Editor Comments:

The paper presents an important investigation into the timing of first intimate partner violence (FIPV) among ever-married women in 30 Sub-Saharan African countries. The utilization of the Gompertz gamma shared frailty model to identify predictors of FIPV is commendable and contributes to the understanding of intimate partner violence (IPV) in low and middle-income countries.

Although the reviewers have accepted the paper, I have couple of concerns below:

The inclusion of variables with p-values less than 0.2 in the bivariable analysis deviates from the standard practice, where only significant variables (p < 0.05) are typically forwarded to the multivariate analysis. I recommend that the authors reanalyse the data, considering only significant variables (p < 0.05) for inclusion in the multivariable analysis.

I am interested to know the clarification on why Gompertz gamma shared frailty modelling was chosen over proportional hazard modelling. Providing justification for this choice in the manuscript would enhance the readers' understanding of the methodology employed in the study.

I recommend major revisions to address the concerns before proceeding with the final decision on the manuscript.

Reviewers' comments:

Reviewer's Responses to Questions

**Comments to the Author**

1. If the authors have adequately addressed your comments raised in a previous round of review and you feel that this manuscript is now acceptable for publication, you may indicate that here to bypass the “Comments to the Author” section, enter your conflict of interest statement in the “Confidential to Editor” section, and submit your "Accept" recommendation.

Reviewer #1: All comments have been addressed

Reviewer #2: All comments have been addressed

2. Is the manuscript technically sound, and do the data support the conclusions?

Reviewer #1: Yes

Reviewer #2: Yes

3. Has the statistical analysis been performed appropriately and rigorously? 

Reviewer #1: Yes

Reviewer #2: Yes

4. Have the authors made all data underlying the findings in their manuscript fully available?

Reviewer #1: Yes

Reviewer #2: Yes

5. Is the manuscript presented in an intelligible fashion and written in standard English?

Reviewer #1: Yes

Reviewer #2: Yes

6. Review Comments to the Author

Reviewer #1: My criticism noted that there was little new in the findings, but I recognise that the topic is of critical importance and there is insufficient attention paid to this major problem of gender violence internationally, but particularly in Africa.

The authors have answered my questions, provided information about the included countries and provided justification for their focus. Reviewer 2 had few criticisms and the authors have justified their use of their methodology. I thus support this publication in Plos One.

Reviewer #2: Intimate partner violence is a nebulous issue that requires different approaches to tackle the menace. This article is a meaningful contribution to the issues around intimate partner violence

7. PLOS authors have the option to publish the peer review history of their article (what does this mean?). If published, this will include your full peer review and any attached files.

Reviewer #1: No

Reviewer #2: **Yes: **Chukwudeh Okechukwu Stephen

---

## [Author Response · Author response to Decision Letter 1]

20 Mar 2024

Point-by-point response 

Point by point response for editors/reviewers comments 

PLOS ONE

Manuscript title: Prognostic factors of first intimate partner violence among ever-married women in Sub-Saharan Africa: Gompertz gamma shared frailty modeling 

Manuscript ID: PONE-D-23-11923R1

 Dear editor/reviewer. 

Dear all,

We would like to thank you for the constructive, building, and improvable comments on this manuscript that would improve the content of the manuscript. We considered each comment and clarification question of editors and reviewers on the manuscript thoroughly. Our point-by-point responses for each comment and question are described in detail on the following pages. Further, the details of changes were shown by track changes in the supplementary document attached

'Response to Editor 

 Editor Comments:

1. Additional Editor Comments:

The paper presents an important investigation into the timing of first intimate partner violence (FIPV) among ever-married women in 30 Sub-Saharan African countries. The utilization of the Gompertz gamma shared frailty model to identify predictors of FIPV is commendable and contributes to the understanding of intimate partner violence (IPV) in low and middle-income countries.

Although the reviewers have accepted the paper, I have couple of concerns below:

The inclusion of variables with p-values less than 0.2 in the bivariable analysis deviates from the standard practice, where only significant variables (p < 0.05) are typically forwarded to the multivariate analysis. I recommend that the authors reanalyze the data, considering only significant variables (p < 0.05) for inclusion in the multivariable analysis.

Author’s response: Dear Academic editor thank you for your comment. We have made some changes to the method section based on the results of our analysis. Although we previously stated that we included variables with a p-value less than 0.2, we have now removed this statement as all variables in the bivariable analysis were statistically significant (p<0.05) with a 95% confidence interval, as shown in Table 5. We have reported the Crude Hazard Ratio (CHR) along with its respective 95% CI.

I am interested to know the clarification on why Gompertz gamma shared frailty modelling was chosen over proportional hazard modelling. Providing justification for this choice in the manuscript would enhance the readers' understanding of the methodology employed in the study.

Author’s response: Frailty models are used to account for unobserved heterogeneity or unmeasured factors that affect the hazard rate in survival analysis. These models introduce a random effect (frailty) that captures individual-specific variations not explained by observed covariates. Shared frailty models assume that the frailties are correlated across individuals within a group (clusters in our case). When the global Schoenfeld residuals test yields a p-value less than 0.05, it indicates that the proportional hazards (PH) assumption has been violated. Consequently, relying solely on a proportional hazard model (PHM) would not be appropriate in this scenario. The violation of the PH assumption implies that the hazard ratios for the covariates are not constant over time. In such cases, alternative survival models that account for time-varying effects or unobserved heterogeneity become necessary. The Gompertz gamma shared frailty model, which incorporates random frailties to capture individual-specific variations, were a suitable choice.

---

## [Editor Report · Decision Letter 2]

22 Apr 2024

Prognostic factors of first intimate partner violence among ever-married women in Sub-Saharan Africa: Gompertz gamma shared frailty modeling

PONE-D-23-11923R2

Dear Dr. Seifu,

We’re pleased to inform you that your manuscript has been judged scientifically suitable for publication and will be formally accepted for publication once it meets all outstanding technical requirements.

Kind regards,

Enamul Kabir

Academic Editor

PLOS ONE
---

## [Editor Report · Acceptance letter]

7 May 2024

PONE-D-23-11923R2 

PLOS ONE

Dear Dr. Seifu, 

I'm pleased to inform you that your manuscript has been deemed suitable for publication in PLOS ONE. Congratulations! Your manuscript is now being handed over to our production team.

Kind regards, 

on behalf of

Dr. Enamul Kabir 

Academic Editor

PLOS ONE